

# The impact of topography on seismic amplification during the 2005 Kashmir Earthquake

Saad Khan[1,2], Mark van der Meijde[2], Harald van der Werff[2], and Muhammad Shafique[3]

[1]Department of Geology, Bacha Khan University Charsadda
[2]Faculty ITC, University of Twente
[3]NCEG, University of Peshawar

**Correspondence:** Saad Khan (saadkhan@geologist.com)

**Abstract.** Ground surface topography influence the spatial distribution of earthquake induced ground shaking. This study shows the influence of topography on seismic amplification during the 2005 Kashmir earthquake. Earth surface topography scatters and reflects seismic waves, which causes spatial variation in seismic response. We perform a 3D simulation of the 2005 Kashmir earthquake in Muzaffarabad with spectral finite element method. The moment tensor solution of the 2005 Kashmir
earthquake is used as the seismic source. Our results show amplification of seismic response on ridges and de-amplification in valleys. It is found that slopes facing away from the source receive an amplified seismic response, and that 98% of the highly damaged areas are located in the topographically amplified seismic response zone.

## 1 Introduction

Intensity and duration of seismic-induced ground shaking is mainly determined by earthquake magnitude, depth of hypocenter,
distance from the epicenter, medium of the seismic waves and site specific geology, topography and regolith (Kramer, 1996; Wills and Clahan, 2006; Shafique and van der Meijde, 2015; Khan et al., 2017). The influence of earth topography on seismic response has been observed and proven numerically and experimentally (Athanasopoulos et al., 1999; Sepúlveda et al., 2005; Lee et al., 2009a). The earth's topography acts as a reflecting surface for upcoming seismic energy and produces surface waves (Lee et al., 2009a, b). The undulating nature of surface topography leads to scattering or focusing of propagating waves (Lee
et al., 2009a, b). Previous studies found that topography amplifies the ground shaking at mountain tops and ridges, while it de-amplifies in valleys; for example Hartzell et al. (1994); Spudich et al. (1996) in California; Lee et al. (2009a, b) in Taiwan; Hough et al. (2010) in Haiti, Kumagai et al. (2011) in Ecuador, and Restrepo et al. (2016) in Colombia. Most seismic active areas are rugged in nature, which makes these regions prone to topographic (de-)amplification (Lee et al., 2009a; Hough et al., 2010; Shafique and van der Meijde, 2015). Incorporating the topographic impact on seismic response is thus important for
seismic hazard assessment, mitigation and seismic shaking prediction (Wu et al., 2008; Bauer et al., 2001; Shafique et al., 2011a).

During the 2005 Kashmir earthquake in northern Pakistan, the city of Muzaffarabad and its surroundings were severely damaged. The earthquake has been studied on various aspects. Ali et al. (2009) studied the impact of surface faults on infrastructure and environment; primarily based on field surveys done immediately after the earthquake. Several satellite based studies pri-



marily focused on field displacement and slip distribution, such as Parsons et al. (2006); Avouac et al. (2006); Pathier et al. (2006); Wang et al. (2007). Others addressed relationships between co-seismic displacement and landslides (e.g. Kamp et al., 2010; Dunning et al., 2007; Saba et al., 2010). Kashmir earthquake induced topographic amplification of seismic responses was first evaluated by Shafique et al. (2008) using the topographic aggravation factor (TAF) after Bouckovalas and Papadim-
itriou (2005). Their method involved the use of topography-derived parameters such as terrain slope and relative height as a proxy for terrain characteristics in a homogeneous halfspace. One of the major simplifications in their work was the use of these pixel-wise proxies instead of a full 3D topographic model. In this paper, we use a 3D spectral element method (SEM) modelling approach that incorporates an elevation model and full elastic waveform simulations, including all possible waves, based on the source characteristics of the earthquake in a homogeneous halfspace.

The SEM was developed by Patera (1984) for computational fluid dynamics, and was introduced for 3D seismic wave prop-agation by (Komatitsch and Vilotte, 1998; Komatitsch and Tromp, 1999). The method is adopted in several studies afterwards; Komatitsch et al. (2004) simulated ground motion in the Los Angeles Basin for the 2001 Mw 4.2 Hollywood earthquake and the 2002 Mw 4.2 Yorba Linda earthquake. Pilz et al. (2011) modeled basin effects on earthquake ground motion in the Santiago de Chile basin using scenario earthquake of Mw 6.0. Magnoni et al. (2014) simulated the 2009 Mw 6.3 L'Aquila Earthquake,
considering topographic and basinal features in central Italy. Lee et al. (2008, 2009b, a, 2013, 2014) developed a real-time online earthquake simulation system based on SEM in Taiwan. Liu et al. (2015) simulated scenario earthquake strong ground motion in the Shidian basin to study basinal influence on seismic amplification and distribution of strong ground motion. Evangelista et al. (2016) studied site response at the Aterno basin (Italy). Paolucci et al. (2016) estimated ground motion for the historical 1915 Marsica earthquake in the Facino basin incorporating topography and bedrock morphology. Restrepo et al.
(2016) simulated 4 scenario earthquakes of Mw 5 along the Romeral fault for the metropolitan area of Medellín (Colombia) demonstrating how topography affects ground response. Smerzini et al. (2017)) studied site effects by taking the historical Mw 6.5 1978 Volvi earthquake in the Thessaloniki urban area (Greece).

In this study we exclusively study the role of topography on ground motion for the area of Muzaffarabad and surrounding areas during the 2005 Kashmir earthquake.

## 2  Study Area

Within the area affected by the 2005 Kashmir earthquake, we selected an area of approximately 40 x 40 km around the city of Muzaffarabad (Fig 1). Being part of the western Himalayas, its position on a converging plate boundary makes this region particularly prone to earthquakes. Its rugged terrain makes it sensitive to topographic (de-)amplification (Lee et al., 2009a; Hough et al., 2010; Shafique and van der Meijde, 2015). The earthquake was caused by reactivation of the Muzaf-
farabad fault (also known as the Balakot-Bagh fault) Hussain et al. (2009) shown in (Fig 1). The Centroid Moment Tensor (CMT) of the (Mw 7.6) 8 October 2005 Kashmir earthquake (Dziewonski et al., 1981; Ekström et al., 2012), retrieved from (http://www.globalcmt.org/) was used for the simulation and lies in the center of the study area (Fig 1). The depth of CMT lies at 12 km. The USGS (https://earthquake.usgs.gov/earthquakes/) reported, after comparing waveform fits based on the two





planes of the input moment tensor (Fig 1), that the nodal plane (strike= 320.0 deg., dip= 29.0 deg.) fits the data better. The seismic moment release based upon this plane is 3.0e+27 dyne.cm and was calculated using a 1D crustal model interpolated from CRUST2.0 Bassin et al. (2000). Several studies (e.g. Avouac et al. (2006); Pathier et al. (2006); Wang et al. (2007) provide detailed information about the fault dynamics, including moment tensor solutions and finite fault models.

## 3  Methodology

We based our analysis on modelling with the spectral element method (SEM) for simulating 3D seismic wave propagation. The software package SPECFEM3D (Computational Infrastructure for Geodynamics, 2016), is used for SEM simulations. SPECFEM3D can simulate global, regional and local seismic wave propagation. It uses the continuous Galerkin spectral-element method to simulate elastic wave propagation caused by earthquakes (Komatitsch and Tromp, 1999).

SEM based modelling relies on meshed objects or volumes. A high-quality 3D mesh is a key factor for a successful application of SEM (Casarotti et al., 2008). A mesh is composed of hexahedra elements that are isomorphous to a cube (Komatitsch et al., 2002). It is defined with material and structural properties that define how it will react to applied conditions (e.g. an earthquake). We used the Cubit v.13.0 software (Sandia National Laboratories, 2011) for generation of the meshes. Surface topography is based on the ASTER Global DEM, a product of National Aeronautics and Space Administration (NASA) and Japan Ministry of Economy, Trade and Industry (METI). It was retrieved from the Global Data Explorer, courtesy of the NASA Land Processes Distributed Active Archive Center (LP DAAC), USGS/Earth Resources Observation and Science (EROS) Center, Sioux Falls, South Dakota, http://gdex.cr.usgs.gov/gdex/.

Previous study has explored at which resolution you can best model the topography in relation to mesh resolution (Khan et al., 2017). They analyzed the impact of data resolution (mesh and DEM) on seismic response using SPECFEM3D. Different combinations of mesh and DEM resolutions were modelled to find the optimal mesh and DEM resolution for getting accurate results while keeping computational resources to the minimum. Their conclusion was that a mesh and topography of 270 m resolution was optimal for the topography around Muzaffarabad. Our model adopts this resolution and thereby allows for seismic wave simulations with frequencies up to ~5.5 Hz. We use a polynomial degree N = 4 to sample the wave field; therefore, each spectral element contains $(N+1)^3 = 125$ Gauss-Lobatto-Legendre (GLL) points, which is 5 GLL points per wavelength. In order to correctly sample the wave field, one needs to use roughly five GLL points per wavelength (Komatitsch et al., 2004). The mesh extends to a depth of 40 km, and contains two tripling layers. Tripling is a refinement technique in meshing for subdividing hexahedral elements in a conforming fashion. Tripling layers increase the spatial resolution of the mesh from 270 m at the surface to 2430 m at the bottom of the model (at a depth of 40 km). This is done in order to reduce the computational time and cost by reducing the total number of mesh elements following the approach as proposed in several other studies (e.g. Lee et al. (2008). Due to the unavailability of a wave speed model for the area, we assigned constant wave speeds (Vp=2800 m/s, Vs=1500 m/s) and density $\rho$=2300 kg/m$^3$) in the modelling, representative for upper crustal conditions (Taborda and Roten, 2015; Wang et al., 2016; Makra and Raptakis, 2016).





To investigate the effect of regional topography on seismic amplification, we ran the 3D model once with a topographic surface and once without topography (having a plain surface instead). As source for the simulation we used the 3D wave field as described in the CMT solution of 2005 Kashmir earthquake. Since SEM is efficient in simulating low frequency ground displacement and has limited capability in simulation of high frequency accelerations (Dhanya et al., 2016) we present our results in peak ground displacement maps (PGD) and PGD ratio maps based on models with and without topography. Based on these PGD maps we created an amplification map. In order to evaluate the impact of topography, the pattern of amplification was compared with the topography of the area. The impact of topography on seismic response was also evaluated by correlating the amplification pattern with the earthquake induced buildings damage and co-seismic landslide data. The damage data (Fig 2) is taken from Shafique et al. (2012), who categorized damage to infrastructure as high, moderate and less. The landslide data (Fig 3) is taken from Humanitarian Information Center Pakistan (HIC-Pakistan), first published by Shafique et al. (2008). Based on the orientation of slopes relative to the CMT location, the aspects of slopes were categorized into away (facing away from the CMT), towards (facing towards the CMT), and other (facing in directions other than towards and away). The categorization is based on 60° set of aspect with respect to its angle towards the CMT location.

This study has been carried out in data sparse environment and therefore we opted to use a homogeneous halfspace model. There is not a single tomographic velocity model available in a relevant resolution. All tomographic velocity models are too coarse; the best resolution is 1 degree spatially with the crust poorly resolved (Johnson and Vincent, 2002). Seismic lines are not available for the region, and nearby seismic lines (Bhukta and Tewari, 2007) cannot provide sufficient detail on the Muzaffarabad region. However, to overcome this simplification, we have tried to establish the velocities as accurate as possible by comparing our results with observed displacements by Pathier et al. (2006); Avouac et al. (2006); Wang et al. (2007). Our displacement values, based on the homogeneous velocity model, are fairly comparable with their results and our velocities are comparable to upper crustal velocities in coarse tomographic models for the region (Johnson and Vincent, 2002; Bhukta and Tewari, 2007). Furthermore, with changing velocities, the absolute values will change but the amplification pattern will still remain the same; for events at the same location the effect of amplification is largely magnitude and velocity independent. Since our model would be, based on previous studies, homogeneous for the upper crust anyway, with no sediments inclusion, the effect of a full homogeneous model is limited. Because of the depth of the earthquake in relation to the limited size of the area the amount of energy that might have been deflected back up is very limited and considered negligible in this study. We are aware that the phenomena exists and is ignored, but any assumption on the velocity model will carry similar uncertainties as to this simplification.

## 4 Methodology

The modelling results show that seismic response is sensitive to slope angle, aspect, geometry, and height of the terrain features. The modelled PGD amplitudes differ for a homogeneous halfspace without topography (Fig 4a) and with topography (Fig 4b). The higher amplitudes coincide with mountain ridges, as shown in the DEM (Fig 4c). Without topography, the PGD falls within the range of 0.23-5.8 m (Fig 4a), but increases to a range of 0.36-7.85 m (Fig 4b) when topography is taken into account. The



difference between the PGDs of the two models (ΔPGD) is shown in Fig 4(d). The difference in PGD between a model with topography versus the same model without topography is represented in terms of amplification. For positive values we use term amplification, meaning the seismic signal has become stronger due to topography compared to simulation without topography. Whereas, for negative values we use de-amplification, meaning the seismic signal has become weaker due to topography

compared to the simulation without topography. The topographic (de-)amplification causes local changes of approximately -2.50 to +3.00 m (Fig 4d).

For a detailed analysis of the effect of topography we compare topography, PGD (with and without topography) and ΔPGD along profile lines. A comparison (Fig 5a, b and c) is made along the white profile lines shown in Fig 4 (AA', BB' and CC', respectively). The profile line AA' is approximately 47.5 km long, and passes over the CMT location in the center of the profile

(marked with a dotted arrow). We observe, in general, amplification at ridges and de-amplification in valleys. The amplification and de-amplification related to ridges and valleys, respectively, has a shift toward the ridge slope facing away from the CMT location (Fig 5a). We find that slopes facing away from the epicenter have an amplified seismic response. Similarly, slopes facing towards the CMT have a de-amplified seismic response. The most clear and prominent example of this amplification is at location (a) in Fig 5(a). The slope facing away from the CMT location is experiencing amplified PGD amplitudes. The

maximum amplitudes occur at the top and near the top on the slope facing away from the CMT location. On the slope side facing towards the CMT location, we see a rapid decrease in amplification, turning into de-amplification for the lower part of the slope. This pattern of decay with elevation we also see for the amplified side but the decay there is much slower and the model shows amplified signals till much further down the slope. Similar patterns can be observed for the ridges at locations (b) and (c) we find that the slope facing towards the source has experienced de-amplification, while the slope facing away

is showing amplification. The same phenomena of amplification and de-amplification can also be observed on slope along profiles BB' and CC' (at locations a, b & c) in Fig 5(b and c) for profile lines BB' and CC', respectively. The trapping of energy due to the shape of the mountain is observed and reflection of energy towards the top of the mountain leads to increased amplification effects with increase in elevation from the base of the mountain, particularly on the side of the mountain that is directly exposed to the incoming seismic waves. A clear shadow effects due to terrain features can be seen at some locations.

At locations (d) and (e) (Fig 5a) we observe that a deep valley blocks the continuation of seismic wave energy into the next topographic high, thereby leading to de-amplification. Similarly, at location (d) in Fig 5(b and c), we observe a similar shadow effect, and resulting de-amplification, due to blocking of seismic waves by deep valleys between the ridge and the CMT.

The ΔPGD shown in Fig 4(d) is compared with damage data (Fig 2) of the 2005 Kashmir earthquake in Fig 6. Shafique et al. (2012) reported damage into three categories; high (red), moderate (yellow) and low (green). The high damage zone constitutes

11% of the damaged part of the Muzaffarabad area, while moderate and low damage respectively cover 51% and 38% of the damaged area. There is clear correlation between (de-)amplification and damage level. For the highest class of damage, 98% of the damaged buildings are found in the amplified zone (positive ΔPGD, Fig 4(d)). On the other hand, 80% of the least damaged buildings lies within the de-amplified zone. Overall the distribution of damage is equally distributed over amplified and de-amplified zones, but amplification does have an impact on the level of damage.



Following the hypothesis that the direction of slopes has an impact on the amplification we would expect that the away facing slopes show a relatively higher ΔPGD than slopes facing toward or any other direction. Analysis of the terrain shows that 30% of the slopes face away from the CMT location, 35% face towards the CMT location, and 35% face another direction (Fig 7). On average we observe that around 2/3 of the area (63%) experiences a de-amplification, and around 1/3 (37%) shows

amplification. When comparing these statistics with the statistics for the different aspect classes a relative increase is observed for slopes that face away from the CMT location to 47%. Contrary, a decrease is observed, to 25%, for slopes facing towards the CMT location. So, the effect of slope direction on the amplification is significant. In line with these observations, there have been studies (e.g. Ashford and Sitar (1997); Ashford et al. (1997)) that suggested a correlation between landslide occurrence and earthquake location as a possible result of amplification (Shafique et al., 2008; Meunier et al., 2008; Qi et al., 2010;

Xu et al., 2013). For the Muzaffarabad area, the ΔPGD was compared with landslide data (Fig 3) collected shortly after the earthquake by HIC-Pakistan Shafique et al. (2008) (Fig 8). This shows a similar pattern as for the previous analysis. On slopes facing away from the CMT location we observe 53% of the landslides in the amplified zone, whereas on slopes facing towards the CMT this is only 16%.

## 5    Discussion

This study used the spectral element modelling numerical modelling to evaluate the impact of topography on 2005 Kashmir earthquake induced ground shaking. Our results show that, by incorporating topography in spectral element modelling, the minimum and maximum amplitude of the peak ground displacement changes. The results show manifestations of topographic influence on building damage during the 2005 Kashmir earthquake. We find that the majority (98%) of the high damaged area lies in the topographically amplified response region. Conversely, the majority (80%) of the area with less damage lies in the

topographically de-amplified response region. The relation between damage and amplification indicates that the topography was a contributing factor to the building damages during the 2005 Kashmir earthquake. It is also important to consider other factors that could play role in damage to infrastructure. Shafique et al. (2011b) has shown that regolith thickness had an influence on damages during the 2005 Kashmir earthquake. Actual damage is also dependent on building quality. Although Asian Development Bank and World Bank (2005) reported that building material and poor construction was homogeneous in

the area, Shafique et al. (2011b) observed differences between different sectors in the area. Poor construction practices such as connected buildings and poor reinforcements, are also considered as contributing factor for damage in the area in response to the 2005 Kashmir earthquake Shafique et al. (2011b) and might have influenced, in a positive or negative sense, the comparison between amplification and damage.

Most of the structures in the area are low-to medium-high, which are normally most sensitive to high-frequency amplification

effects. However, considering such high PGDs in this area, combined with building style not earthquake proof, the relation between PGD and damage might not be optimal but is thought to show a strong correlation. Furthermore, during this study, PGD, peak ground velocity (PGV) and peak ground acceleration (PGA) have been found spatially very strongly correlated,



with only some minor amplitude deviations at specific topographic features. So the overall pattern in a comparison would still look very similar to the damage vs ΔPGD comparison we show in the paper.

The directions of incident seismic waves have a significant impact on distribution of seismic-induced landslides (Ashford and Sitar, 1997; Shafique et al., 2008; Meunier et al., 2008; Qi et al., 2010; Xu et al., 2013). In our results this effect is not

directly clear. For example, only 37% of the landslides fall within the amplified zone, the other 63% fall within the de-amplified zone (Fig 8). However, when the relative distribution of landslides is compared to slopes facing away (prone to amplification) versus slopes facing towards the CMT location, a correlation is observed. The majority of the landslides on slopes facing away from the CMT location are in the amplified zone (53%) while for slopes facing towards the CMT location this is only 16%. These figures are much lower than Shafique et al. (2008) results for the same area and the same landslide catalogue. The main

difference between their study and ours is the location to which the slope direction is compared. While Shafique et al. (2008) used the onset of the earthquake (the Epicenter location, Fig 1), we used in our modelling the CMT location (the location of maximum energy release, Fig 1). It was assumed that the maximum seismic amplitudes and corresponding amplification will lead to the occurrence of landslides but apparently this is not true. Based on the comparison of the two results it is very likely that a much lower displacement value may have already triggered the landslides. The threshold at which the landslides

will occur cannot be derived from this study but it is evident that already at much earlier stage in the earthquake propagation enough energy is created to activate landslides. The maximum energy release related to the CMT location occurs only later in the earthquake process and clearly has less control since the critical ground displacement required for activating a landslide might have already been exceeded. This can explain the abundance of landslides in the de-amplified zone based on modelling from the CMT location.

The aforementioned evidence of amplified seismic response on slopes facing away from the source could be a possible reason behind triggering landslides. However, it is important to consider complications associated with landslides. The 2005 Kashmir earthquake induced landslides have been addressed in several studies, from different perspectives. Owen et al. (2008) reported that more than half of the landslides were in some way associated with road construction and other human activity. According to Kamp et al. (2008), bedrock lithology (comprising highly fractured slate, shale, dolomite, limestone and clastic

sediments) was the most important landslide controlling parameter during the 2005 Kashmir earthquake. Dellow et al. (2007) reported a highly asymmetric size and distribution of 2005 Kashmir earthquake induced landslides. According to this last paper, the landslides can be classified into the following three types: (1) landslides formed over/adjacent to the fault rupture, (2) landslides which extend about 10–20 km from the fault trace on the hanging side of the fault and (3) landslides on the footwall side which are generally rare except within 2–3 km of the fault trace. And as mentioned earlier, Shafique et al. (2008)

compared the aspect of landslides relative to the initial rupture point and found that about 80% of the landslides had an aspect facing away from the rupture point. In summary, the major controlling factors for 2005 Kashmir earthquake induced landslides were (1) human activity (Owen et al., 2008), (2) bedrock lithology (Kamp et al., 2008), (3) proximity with reference to fault trace (Dellow et al., 2007) and (4) slope direction with respect to source (Shafique et al., 2008). In our study, we analyzed landslide aspects with respect to the point of maximum release of energy (CMT solution). The percentage of landslides facing

away was found to be 25%, facing towards 33% and facing other directions 42%. Keeping in mind factors such as human





activity and bedrock lithology beside the relation of landslides with the fault trace, it is uncertain at which stage of the rupture the landslides have been triggered. It could be because of the initial rupture (used by Shafique et al. (2008), the fault trace (Owen et al., 2008), the moment of maximum release of energy (the CMT solution location used in this study), or somewhere in between.

The 2005 Kashmir earthquake was a shallow earthquake. In such case a seismic wave field will reach the surface at an angle, rather than vertical when originating at a larger distance away. This can lead to the creation of a so-called shadow zone effect due to deep valley blocking the propagation of a seismic wave field into a topographic feature. This phenomenon has also been observed in this study. Because of this shadow effect, ridges which are expected to show amplification, show de-amplification (location (d) & (e) in Fig 5(a), location (d) in Fig 5(a and b). This effect however, may not be visible for deep seated seismic

sources or sources at a larger epicentral distance. The deeper or further away the source of the earthquake the more vertical will be the incoming seismic wave field. A similar effect is possible if we have significantly reduced seismic velocities close to the surface, due to e.g. sediments, that will also turn the wave field towards the vertical. The results of the study can be used as an important parameter for the seismic microzonation for the study area to mitigate the negative impacts of earthquake.

## 6  Conclusions

Topography affects the diffraction and reflection of incident seismic waves, thereby amplifying or de-amplifying the seismic response. The impact of topography on seismic induced ground shaking was evaluated for the 2005 Kashmir earthquake in Muzaffarabad (Pakistan) using a spectral element method, SPECFEM3D. An ASTER Global DEM, re-sampled to 270 m spatial resolution, was used for representing topography of the study area. A mesh of 270 m spatial resolution was used to model the topography and geometry of the topography and subsurface conditions in the Muzaffarabad region. Overall,

topography induced amplification of seismic response is found on ridges and slopes facing away from the CMT location and de-amplification is found in valleys and at the bottom of slopes facing towards the CMT location, which is consistent with previous studies. The study demonstrates that topography changed the PGD values from approximately -2.5 m to +3 m when compared to a plain mesh surface.

    It is shown that topography played a significant role in earthquake induced damage during the 2005 Kashmir earthquake:

98% of the highly damaged area lies within the topographically amplified seismic response area.

*Competing interests.*  No competing interests are present.

*Acknowledgements.*  The damage data was obtained from Shafique et al. (2012) and is based on a post-earthquake SPOT5-image combined with field observations. The landslide inventory was developed by the Humanitarian Information Center (HIC), a subsidiary organization of the United Nations Office for the Coordination of Humanitarian Affairs (OCHA).

Funding: The research budget was provided by ITC, University of Twente via OFI No. 93003030.



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





**Figure 1.** Digital Elevation Model (DEM) of the study area of Muzaffarabad (Pakistan). The Main Boundary Thrust (MBT) and its segments (after Hussain et al. (2009)) are shown along with Centroid Moment Tensor (CMT) solution.





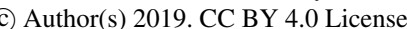

**Figure 2.** 2005 Kashmir earthquake induced damage data after Shafique et al. (2012).



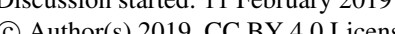

**Figure 3.** Co-seismic landslides inventory was developed by the Humanitarian Information Center (HIC), a subsidiary organization of the United Nations Office for Coordination of Humanitarian Affairs (OCHA), which was reported by Shafique et al. (2008).







**Figure 4.** (a) Peak Ground Displacement (PGD) for model without topography, (b) PGD for model with topography, (c) topography (ASTER Global DEM at 270 m), (d) difference between (a) and (b) (ΔPGD). The red tones (positive values) indicate amplification, blue tones (negative values) indicate de-amplification, while the green represents zero difference. The CMT location is represent by red beachball.





**Figure 5.** Profile along line (a) AA', (b) BB' and (c) CC' (Fig 4) of PGD with topography (dotted green) and without topography (dotted olive), their difference (ΔPGD, solid red) and elevation (solid blue). PGD are scaled on the left y-axis along line, elevation is shown on the right y-axis. The profile line passes over the CMT location which is located approximately in the center of the profile, marked with a dotted arrow line.





| Damage | De-amplified zone | Amplified zone | Spatial Extent |
|---|---|---|---|
| Less (■) = | 80 % | 20 % | 38 % |
| Moderate (▫) = | 46 % | 54 % | 51 % |
| High (▪) = | 02 % | 98 % | 11 % |
| Total (▪+■+▫)= | 54 % | 46 % | 100 % |

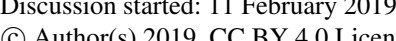

**Figure 6.** The difference in peak ground displacement (ΔPGD) between models with and without topography (Fig 4d) compared with damage data of Shafique et al. (2012). The top part shows the relation between damaged areas with (de-)amplification in table form. The bottom part shows the distribution of damage with ΔPGD graphically. The high damage zone constitutes 11% of the Muzaffarabad damaged area, while moderate and low damage respectively cover 51% and 38% of the damaged area.

| | De-amplified zone | Amplified zone | Spatial Extent |
|---|---|---|---|
| Away facing (▪) = | 53 % | 47 % | 30 % |
| Towards facing (■) = | 75 % | 25 % | 35 % |
| Other facing (▫) = | 58 % | 42 % | 35 % |
| Total (▪+■+▫)= | 63 % | 37 % | 100 % |

**Figure 7.** The difference in peak ground displacement (ΔPGD) between model with and without topography (Fig 4d) is compared with the aspects of the study area. The aspects are categorized into away (facing away from epicenter), towards (facing epicenter), and other (facing a direction other than towards and away). The categorization is based on 60° offset of aspect with respect to its angle towards the CMT.

| Aspect direction | De-amplified zone | Amplified zone | No. of Landslides (%) |
|---|---|---|---|
| Away facing (▪) = | 47 % | 53 % | 25 % |
| Towards facing (■) = | 84 % | 16 % | 33 % |
| Other facing (▫) = | 56 % | 44 % | 42 % |
| Total (▪+■+▫)= | 63 % | 37 % | 100 % |

**Figure 8.** The difference in peak ground displacement (ΔPGD) between the models with and without topography (Fig 4d) is compared with the aspects of earthquake induced landslides (Source: Humanitarian Information Center Pakistan (HIC-Pakistan), by courtesy of Shafique et al. (2008). The aspects are categorized into away (facing away from epicenter), towards (facing epicenter), and other (facing a direction other than towards and away). The categorization is based on 60° offset of aspect with respect to its angle towards the CMT.