# Peer review of "The impact of topography on seismic amplification during the 2005 Kashmir Earthquake"

_Natural Hazards and Earth System Sciences, 2019_

## Referee Comment (RC1) · Steven de Jong (Referee) · 25 Feb 2019

General: The study presented in this paper shows an interesting case study of modelling the effect of topography on local ground shaking during the Kashmir earthquake. Although I am a bit away from the scientific field and not an expert in SEM - SPECFEM3D I believe the study in an interesting contribution to our knowledge on seismic amplification and the role of topography, applied to an interesting study area of the Muzaffarabad fault and to a relevant event. The presented discussion (pages 6 and 7) on the relation between seismic amplification and the occurrence and patterns of landslides is very interesting. I understood this is already a revised version of an original paper. With some minor adjustments I think it can be accepted.
Specific Comments: The approach chosen to evaluate the effect of regional topography to run scenarios with and without topography, to separate low and high frequency accelerations as described on page 4 top seems a valid approach. On page 4 line 14 is mentioned that a 'homogenous halfspace model is used. This might need some clarification what that means in the model and what the implication are for the outcome of the model and accuracies. A mesh size of 270 m was chosen and used. Not sure what the motivation for this meshsize is. Please add a few words on your considerations.

Technical: Sections 3 and 4 are both titled 'Methodology' in my version of the paper. Section 4 must read Results I suppose. Figures are relevant and well taken care of.

---

## Referee Comment (RC2) · Anonymous Referee #2 · 25 Jul 2019

This is a review on the paper entitled "The impact of topography on seismic amplification during the 2005 Kashmir earthquake" by Saad Khan et al. The paper is interesting, valuable, and well organized. Although I am tending to accept the paper, the following points should be addressed before publication. 1- Authors calculated the topographic effect using 3-D model once with topographic effect and once without topographic effect. The topographic effect should include the effects due to the present valleys on the ground motion, thus the selected plain surface should be free of the valleys' effects. Authors should provide the characteristics of their selected datum. 2- The paper lacks the description of both depth of the earthquake and depth of the valleys to be sure that these valleys are really shadow zones preventing seismic waves from reaching high areas on the other sides. Detailed description of low areas is required 3- Many factors

can amplify ground motion. To have accurate correlation between the topography and the observed damage in the region, all other factors should be neutralized in advance to be sure about the effectiveness of topographical contribution. This is not clear in the current manuscript. 4- It seems that authors modeled the seismic source as a point, which is totally unreliable, as the rupture direction could be an effective parameter at short distances. Details on the fault rupture direction, rupture length, and observed surface displacement should be provided. 5- As the earthquake is relatively recent, field observations of such earthquake should be available. Therefore, verification of the calculated values with the recorded observation should be provided to be sure about the accuracy of the used model (including input uncertainty) and the results. Numerical modelling alone is not enough. Minor comments a) Use the past tense in the abstract section. b) Rewrite line No. 10 in page 1, modifying the position of the word "and" and removing the word regolith as it is a part of the site specific geology. c) Line 20 page 1. Seismic risk cannot be mitigated. Use risk instead. d) Page 3, line 9, elastic waves. e) Page 3, lines 30 and 31, use velocity instead of speed. f) Title of section 4 should be Results. g) Page 5, lines 12 and 13, give possible reasons. h) Page 6, line 18, found instead of find.

---

## Author Comment (AC1) · 19 Oct 2019

We are thankful for the reviewer's comments and have taken care of addressing all of them. In the attached zip file you will find a pdf replying to each comment individually, and references to the location in the manuscript (also in pdf) where the suggestions have been incorporated (underlined where changes/improvements are made).

Please also note the supplement to this comment:
https://www.nat-hazards-earth-syst-sci-discuss.net/nhess-2019-13/nhess-2019-13-AC1-supplement.zip
* * *
[Figure]

2019-13, 2019.

---

## Author Response (AR1)

**The impact of topography on seismic amplification during the 2005 Kashmir Earthquake**

**Saad Khan*[1,2], Mark van der Meijde[2], Harald van der Werff[2], Muhammad Shafique[3]**

[1]Department of Geology, Bacha Khan University Charsadda
[2]Faculty of Geo-information and Earth Obsevation (ITC), University of Twente
[3]National Center of Excellence in Geology (NCEG), University of Peshawar

*Correspondence email: saadkhan@bkuc.edu.pk

We are thankful for the reviewer's comments and have taken care of addressing all of them. Below you will find replies to each comment individually, and references to the location in the manuscript where the suggestions have been incorporated (underlined where changes/improvements are made).

| Reviewer 1 | | |
|---|---|---|
| **Comment** | **Reply** | **Manuscript Reference** |
| The approach chosen to evaluate the effect of regional topography to run scenarios with and without topography, to separate low and high frequency accelerations as described on page 4 top seems a valid approach. On page 4 line 14 is mentioned that a 'homogenous halfspace model is used. This might need some clarification what that means in the model and what the implication are for the outcome of the model and accuracies. | A halfspace is a simplified mathematical model used to approximate the earth when performing seismological calculations. In a homogeneous halfspace, material/velocity properties are kept constant throughout the model. There are mainly two reasons behind adopting homogeneous halfspace instead of a heterogeneous halfspace (where the material/velocity properties changes).

 1. Non-availability of the tomographic velocity model.
 2. To avoid any effect of heterogeneity on amplification.

 Non-availability of the tomographic velocity model, especially at the resolution adopted in this study. This may change the absolute ground motion values but not the amplification due to topography (except when there are sediments, which we have explicitly excluded in our modeling due to large uncertainties in the sediment thickness in the area and the overprint on a possible topographic seismic amplification effect). It might have a slight effect on the spatial amplification pattern; an incoming wavefield can come in under a different angle if a layered velocity model will be used, but we have estimated that the effect of making a guess for the correct global velocity model for an intra-crustal earthquake is | Lines 19-35 on page 4. Lines 1-2 on page 5. |

| | | |
|---|---|---|
| | as uncertain as the choice for a homogeneous upper crustal velocity model.
This is discussed in the last paragraph of the methodology section. Based on this comment, we decided to revise the manuscript to make this explanation clearer. | |
| A mesh size of 270 m was chosen and used. Not sure what the motivation for this meshsize is. Please add a few words on your considerations. | The motivation behind this choice was briefly discussed in the Methodology section and is based on previous research (Khan et al., 2017). In this earlier paper we have tested several mesh resolutions to find the best approach in a trade-off between accurate results and computing time. It was shown that for the geomorphological geometry for Pakistan a 270 m mesh resolution gives accurate results and a significant decrease in accuracy is observed for coarser models. Based on the question, we decided to revise the text for better clarity. | Lines 17-23 on page 3. |
| Sections 3 and 4 are both titled 'Methodology' in my version of the paper.
Section 4 must read Results I suppose. | Thank you; we changed that. | Line 3 on page 5. |

**The impact of topography on seismic amplification during the 2005 Kashmir Earthquake**

Saad Khan*[1,2], Mark van der Meijde[2], Harald van der Werff[2], Muhammad Shafique[3]

[1]Department of Geology, Bacha Khan University Charsadda
[2]Faculty of Geo-information and Earth Observation (ITC), University of Twente
[3]National Center of Excellence in Geology (NCEG), University of Peshawar

*Correspondence email: saadkhan@bkuc.edu.pk

We are thankful for the reviewer's comments and have taken care of addressing all of them. Below you will find replies to each comment individually, and references to the locations in the manuscript where the suggestions have been incorporated (underlined where changes/improvements are made).

| Reviewer 2 | | |
| --- | --- | --- |
| **Comment (Major)** | **Reply** | **Manuscript Reference** |
| Authors calculated the topographic effect using 3-D model once with topographic effect and once without topographic effect. The topographic effect should include the effects due to the present valleys on the ground motion, thus the selected plain surface should be free of the valleys' effects. Authors should provide the characteristics of their selected datum. | We consider the zero elevation surface to be the datum. It is sampled with a 270 m mesh resolution and DEM resolution. This zero elevation surface removes any impact of valleys.

It should be noted that a choice for any other datum (e.g. the valley bottom) would actually give the same output, as long as that the reference datum is below the (deepest) valley floor to ensure that all topography and geomorphological characteristics are included in the model with topography and excluded in the model without.

This additional information about the datum has been added to the manuscript for clarification. | Line 34 on page 3.
Lines 1-3 on page 4. |
| The paper lacks the description of both depth of the earthquake and depth of the valleys to be sure that these valleys are really shadow zones preventing seismic waves from reaching high areas on the other sides. Detailed description of low areas is required. | We agree that this can be explained in more detail, the present description can possibly lead to confusion. Information about the depth of earthquake and deep valleys causing shadow effects is now provided in the manuscript. | Lines 31 on page 2.
Lines 31-32 on page 5.
Lines 1-2 on page 6.
Lines 27-31 on page 8. |
| Many factors can amplify ground motion. To have accurate correlation between the topography and the observed damage in the | We agree with this comment; it is now better clarified in the manuscript. | We added an explanation to lines 25-32 on page 6 and lines 1-2 on page 7. |

| | | |
|---|---|---|
| region, all other factors should be neutralized in advance to be sure about the effectiveness of topographical contribution. This is not clear in the current manuscript. | | |
| It seems that authors modeled the seismic source as a point, which is totally unreliable, as the rupture direction could be an effective parameter at short distances. Details on the fault rupture direction, rupture length, and observed surface displacement should be provided. | Details on the fault rupture direction, rupture length, and observed surface displacement has been provided along with justification for using a point source in the manuscript. Previous studies (e.g. Raghukanth, 2008) have shown that there is a strong correlation with the CMT location (i.e. point of maximum energy release) and damage patterns. We also believe that this maximum energy release, and thereby the max PGA/PGV/PGD will be the dominant trigger for the occurrence of landslides, so amplification in relation to CMT location is assumed to be the most dominant direction for amplification related damage and secondary hazards.
We have added text on this issue in the discussion session of the paper. | Lines 4-9 on page 4.
Lines 30-32 on page 6. |
| As the earthquake is relatively recent, field observations of such earthquake should be available. Therefore, verification of the calculated values with the recorded observation should be provided to be sure about the accuracy of the used model (including input uncertainty) and the results. Numerical modelling alone is not enough. | Unfortunately, the region has a poorly developed seismic network and therefore we do not have any recorded seismic data to compare with. The only verifiable data from the field we could access is the damage data (Shafique 2012) and landslide data (Shafique 2008) collected right after the event (E.g. Figure 1 and Figure 2 here in this document); Apart from this, there are some COSI-Corr (Leprince et al., 2008; Avouac et al., 2006) and InSAR based studies (Pathier et al., 2006; Wang et al., 2007) to compare with. COSI-Corr can only measure horizontal component while InSAR only vertical component of displacement. All these comparisons have been discussed and cited in the manuscript, and we have found that our amplitudes are of the same order as theirs.

Any limitations resulting from this drawback are, however, clearly discussed in detail in discussion section. We realize that this leaves maybe some discussion on the final interpretation, which we extensively describe in the discussion section. But on the other | Lines 20-35 on page 4.
Lines 1-2 on page 5.
Lines 7-20 on page 7.
Lines 27-35 on page 7.
Lines 1-22 on page 8. |

| | hand, this is the only possible way to understand better the effects of earthquakes in remote areas. | |
|---|---|---|
| **Comment (Minor)** | | |
| Use the past tense in the abstract section. | Done | Abstract |
| Rewrite line No. 10 in page 1, modifying the position of the word "and" and removing the word regolith as it is a part of the site specific geology. | Done | Line 10 on page 1. |
| Line 20 page 1. Seismic risk cannot be mitigated. Use risk instead. | You are right, we changed the phrasing to: *Incorporating the topographic impact on seismic response is thus important for seismic shaking prediction, seismic hazard assessment and risk mitigation.* | Changed in line 19-20 on page 1. |
| Page 3, line 9, elastic waves. | Done | Line 8 on page 3. |
| Page 3, lines 30 and 31, use velocity instead of speed. | Done | Lines 29-30 on page 3. |
| Title of section 4 should be Results. | Done | Line 3 on page 5. |
| Page 5, lines 12 and 13, give possible reasons. | Done | Line 25 on page 7. |
| Page 6, line 18, found instead of find. | Done | Line 5 on page 7. |

[Figure]

Figure 1: Impact of topography on building damages during 2005 Kashmir earthquake in Balakot (Pakistan). It can be observed that the building on ridge are completely destroyed while those at the ridge toe are still intact, despite the fact that building material and construction was similar.

[Figure]

Figure 2: Impact of topography on building damages during 2005 Kashmir earthquake in Muzaffarabad (Pakistan). Same observation as in Figure 1.

[revised manuscript text omitted]